# The landscape of alternative polyadenylation in single cells of the developing mouse embryo

Vikram Agarwal [1,6 ✉], Sereno Lopez-Darwin[2,6], David R. Kelley [1] & Jay Shendure [2,3,4,5 ✉]

3′ untranslated regions (3′ UTRs) post-transcriptionally regulate mRNA stability, localization, and translation rate. While 3′-UTR isoforms have been globally quantified in limited cell types using bulk measurements, their differential usage among cell types during mammalian development remains poorly characterized. In this study, we examine a dataset comprising ~2 million nuclei spanning E9.5–E13.5 of mouse embryonic development to quantify transcriptome-wide changes in alternative polyadenylation (APA). We observe a global lengthening of 3′ UTRs across embryonic stages in all cell types, although we detect shorter 3′ UTRs in hematopoietic lineages and longer 3′ UTRs in neuronal cell types within each stage. An analysis of RNA-binding protein (RBP) dynamics identifies ELAV-like family members, which are concomitantly induced in neuronal lineages and developmental stages experiencing 3′-UTR lengthening, as putative regulators of APA. By measuring 3′-UTR isoforms in an expansive single cell dataset, our work provides a transcriptome-wide and organism-wide map of the dynamic landscape of alternative polyadenylation during mammalian organogenesis.

[1] Calico Life Sciences, South San Francisco, CA, USA. [2] Department of Genome Sciences, University of Washington, Seattle, WA, USA. [3] Howard Hughes Medical Institute, Seattle, WA, USA. [4] Brotman Baty Institute for Precision Medicine, University of Washington, Seattle, WA, USA. [5] Allen Discovery Center for Cell Lineage Tracing, Seattle, WA, USA. [6] These authors contributed equally: Vikram Agarwal, Sereno Lopez-Darwin. ✉email: vagar@calicolabs.com; shendure@uw.edu

During transcriptional elongation, the cleavage and polyadenylation machinery governs the specification of the 3′-terminal end of an mRNA[1]. This regulated process can generate a diversity of 3′-UTR isoforms for any given gene, dramatically altering the 3′-UTR length and sequence of the resulting mature transcripts[2]. This phenomenon, known as alternative polyadenylation (APA), has been observed in over 70% of mammalian genes[3,4]. Alternative 3′-UTR isoforms bind to different sets of microRNAs and RNA-binding proteins, which collectively modulate a multitude of post-transcriptional gene regulatory mechanisms[5]. These include changes in mRNA localization[6], degradation rates[7–9], and translational efficiency[10]. The differential abundance of a variety of nuclear factors serves to regulate APA in a cell-type-specific manner[1]. Abnormal regulation of the cleavage and polyadenylation machinery has also been associated with hyperproliferative or disease states such as cancer[11–13].

Techniques to directly measure APA in the transcriptome largely rely upon the isolation of RNA from bulk tissue, resulting in an average readout of the landscape of 3′-ends in a heterogeneous population of cells. Existing 3′-end sequencing methods include 3′-seq/3SEQ[14,15], 3P-seq[8,16], PAS-seq[17], 3′READS[18], PolyA-seq[3], and 2P-seq[19]. The successful application of these methodologies in mammalian cells has led to the annotation of hundreds of thousands of polyadenylation sites (PAS) in both human and mouse genomes[20,21]. Bulk 3′-end sequencing and similar transcriptomic data have guided the observation that 3′-UTRs generally lengthen during mammalian embryogenesis[22], with proliferating cell types such as blood exhibiting shortened 3′-UTRs[11,12] and neuronal ones exhibiting lengthened 3′-UTRs[17,23].

In contrast to bulk methods, single-cell RNA sequencing (scRNA-seq) protocols capture a rich diversity of individual cell types, with many protocols enriching for mRNA 3′-ends via poly(A) priming[24–29]. Thus, these technologies inherently offer an unprecedented opportunity to observe APA events during the process of cellular differentiation. They also enable the decomposition of complex tissues into individual cell types, enabling the assessment of APA with greater cell-type resolution. Although a proof-of-concept study has demonstrated the utility of scRNA-seq data in evaluating APA[30], such methods have not yet been applied to investigate more comprehensive scRNA-seq datasets such as those capturing dozens of cell types during a mammalian developmental time course[31,32]. In this study, we examined APA using MOCA ("mammalian organogenesis cell atlas"), a dataset comprising single nucleus transcriptional profiling of ~2 million nuclei encompassing 38 major cell types across five stages (i.e., E9.5, E10.5, E11.5, E12.5, and E13.5) of mouse embryonic development[31].

## Results

**An integrated annotation set of 3′-UTRs and poly(A) sites to evaluate APA.** Given the reliance of many scRNA-seq protocols on poly(A) priming, such methods enrich for both mRNA 3′-ends as well as internal A-rich stretches of homopolymers. Thus, internal priming artifacts obscure accurate quantitation of APA, even more so in datasets in which immature mRNAs (i.e., without excised introns) are isolated from the nucleus, as is the case with the sci-RNA-seq3 protocol used in MOCA[31]. Please note that in the remainder of the manuscript, we often use the term "cells" in relation to scRNA-seq profiles. However, all MOCA data were derived from nuclei rather than cells. To address the source of bias emerging from internal priming artifacts, we sought to develop a simple computational method to deconvolve the data to specifically isolate and quantify mRNA 3′-ends. Toward this goal, we built integrated databases of poly(A) site (PAS) and 3′-UTR

annotations to guide the interpretation of which subset of mapped reads were supported by orthogonal evidence to reflect authentic 3′-termini, as opposed to A-rich sites internal to a mature or nascent transcript. In doing so, our goal was to minimize the shortcomings of any individual database, each of which utilizes different data sources and strategies for PAS and 3′-UTR annotation.

To generate a reliable PAS set, we considered three of the most comprehensive mouse PAS annotation databases available with respect to the mm10 mouse genome build: Gencode M25[33], which contains 56,592 PASs; PolyA_DB v3[20], which contains 128,052 PASs; and PolyASite 2.0[21], which contains 108,938 PASs. The three databases differ in their use of manually curated annotation and Expressed Sequence Tag (EST) data (as in Gencode M25[33]), the amount of 3′-end sequencing data (246 and 178 mouse samples for PolyA_DB v3[20] and PolyASite 2.0[21], respectively), and their computational processing pipelines. We intersected the PASs from each pair of these resources to evaluate the consistency among databases. While the majority of sites were present in at least two databases, 40.0%, 29.4%, and 30.4% were unique to PolyA_DB, PolyASite, and Gencode, respectively (Fig. 1a). To verify the reliability of PASs present in only a single database (and therefore the most likely to contain false positives), we plotted the profile of nucleotide frequencies in the ± 50 nt region surrounding the annotated cleavage and polyadenylation sites (Supplementary Fig. 1a). The unique PASs of each resource exhibited profiles consistent with positionally enriched mammalian motifs known to guide mRNA cleavage, including several U-rich motifs, the upstream AAUAAA motif, and the downstream GU-rich motif[34]. Moreover, we detected strong enrichment of reads mapping immediately upstream of this set of PASs, with the strongest enrichment spanning the −300 nt to +20 nt region around the PAS (Supplementary Fig. 1b). Given that each PAS database was enriched in known PAS motifs, associated with mapped reads, and held information complementary to the other databases, we carried forward an integrated PAS set derived from the union of the three databases. This integrated PAS set recapitulated these same characteristics, exhibiting both consistency with known PAS motifs and strong read enrichment upstream of the sites (Fig. 1b, c). Finally, to link PASs to specific genes, we utilized our previous 3′-UTR annotation pipeline ("Methods")[9,35] to establish an integrated set by carrying forward the longest 3′-UTRs from four resources: (i) Gencode M25[33], (ii) RefSeq[36], (iii) 3′-UTRs with extreme lengthening[23], and (iv) bulk 3P-seq-based annotations derived from ten mouse tissues and cell lines[8]. This integrated 3′-UTR annotation set helped minimize the possibility that a PAS may be annotated outside of a known 3′-UTR and thus remain unlinked to a specific gene.

Using our integrated PAS and 3′-UTR databases (Fig. 1d), we sequentially filtered our scRNA-seq reads from MOCA to focus on the subset mapping to 3′-UTRs within the −300 to +20 vicinity of a known PAS (Fig. 1c). Due to the abundant mapping of reads to introns in the nucleus-derived MOCA dataset (Supplementary Fig. 1c), generally representing internal priming within unspliced transcripts, there was nearly a tenfold loss in read counts after iterative steps of filtering; however, over 200 million reads were carried forward (Supplementary Fig. 1d). As expected, discarded reads were associated with enriched A-richness downstream (but not upstream) of their 3′-termini (Supplementary Fig. 1e), and were not significantly associated with cryptic PASs absent from our unified PAS database (Supplementary Fig. 1f). Next, we counted reads passing the filtering steps towards the single annotated PAS in its vicinity, enabling the tabulation of read counts associated with each PAS (Fig. 1d). In ambiguous cases in which a read was located in the

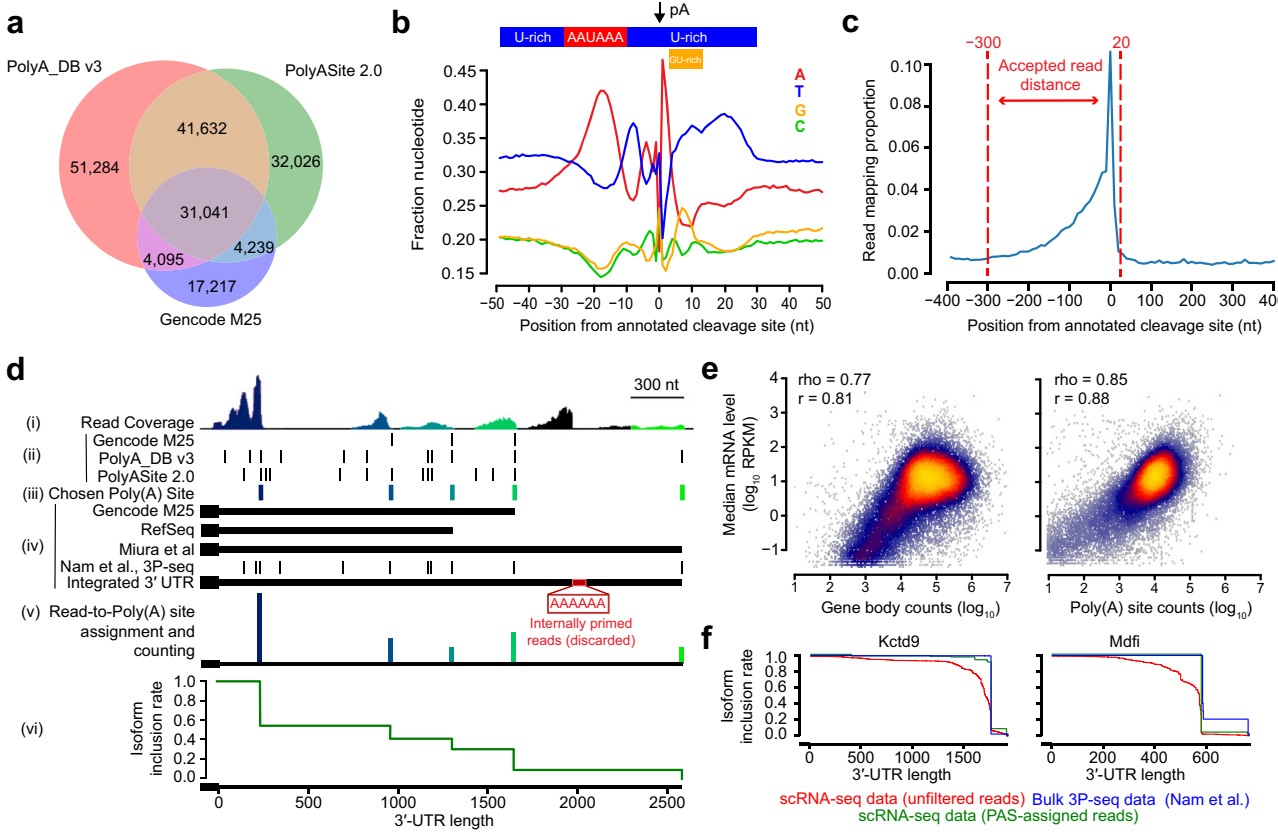

**Fig. 1 A computational pipeline to accurately quantify 3′-UTR isoform abundances from scRNA-seq data. a** Venn diagram of a set of three PAS annotation resources and their degree of intersection. A PAS intersecting within ± 20 nt from another was considered an intersecting hit to account for the heterogeneity of the cleavage and polyadenylation machinery[16]. **b** Profile of nucleotide frequencies in the ± 50 nt vicinity of the annotated cleavage site position, derived from the union of the three databases. Shown above the plot are the known positionally enriched mammalian motifs known to guide mRNA cleavage[34]. **c** Distribution of scRNA-seq reads mapping within the ± 400-nt vicinity of the annotated cleavage site position, derived from the union of the three databases. To avoid an ambiguous signal, the analysis was restricted to PASs not within the same ± 400-nt window as another PAS. Data are binned at 5-nt resolution. Shown within the dotted red lines are the acceptable distance thresholds to associate a read to an annotated PAS. See also Supplementary Fig. 1 for comparisons of (**b**, **c**) for each individual PAS database. **d** Schematic depicting the association of each scRNA-seq read to a PAS in order to quantify relative PAS abundances for a gene. Shown from top to bottom are: (i) the read coverage of scRNA-seq reads mapped to the gene. (ii) The three PAS annotation resources considered, showing the location of each PAS along the 3′-UTR. (iii) The subset of chosen PASs to which reads were greedily assigned, colored from blue to green to indicate which reads from the coverage plot were assigned to them. (iv) The three gene annotation databases integrated with bulk 3P-seq data from ten tissues and cell lines[8] to identify the longest known 3′-UTR. This integrated 3′-UTR was used to associate PASs to the gene. (v) A visualization of relative 3′-UTR isoform abundances after read-to-PAS assignment, with vertical lines at each chosen PAS proportional to the assigned number of read counts. Reads not overlapping within the −300 to +20 vicinity of a known PAS were treated as likely internal priming artifacts and discarded. (vi) The resulting isoform inclusion rate (IIR) plot to quantify the cumulative proportion of 3′-UTR isoforms remaining along the length of a 3′-UTR. See also Supplementary Data 1 for the integrated 3′-UTR database and gene annotations. **e** Scatter plots comparing gene expression levels estimated using scRNA-seq read abundances mapping to the full gene body (left panel) or the sum of reads mapping to PASs (right panel), relative to median gene expression levels from bulk RNA-seq data[37] (n = 19,517 protein-coding genes). Regions are colored according to the density of data from light blue (low density) to yellow (high density). Shown are the corresponding Pearson (r) and Spearman (rho) correlations for each comparison. See also Supplementary Fig. 2 for sequence features explaining biased estimates in the gene body approach. **f** Shown are IIR plots for two genes, comparing the profiles for the raw scRNA-seq data and post-processed data after read-to-PAS assignment with respect to the profile for bulk 3P-seq data[8] as a gold standard. Slight vertical jitter was added for enhanced line visibility. See also Supplementary Fig. 3 for a global comparison among all genes.

vicinity of multiple PASs, we greedily assigned the read to count towards the PAS harboring the most uniquely assignable reads. Finally, based upon the relative counts assigned to each PAS for a given gene, we visualized the "isoform inclusion rate" (IIR), reflecting the proportion of 3′-UTR isoforms which include a given nucleotide position[8,9,35] (Fig. 1d).

To validate that these filtering and read-to-PAS assignment procedures led to reliable results, we performed two quality control (QC) comparisons. As a first QC, reasoning that the removal of internal priming artifacts should improve the quantification of relative gene expression levels, we compared

the relationship between PAS counts and median gene expression levels computed across a panel of 254 mice RNA-seq samples[37]. While the traditional method of counting reads in the gene body displayed a strong correlation to median expression levels (Pearson r = 0.81, Spearman rho = 0.77), it displayed a clear bias in inflating estimates for a large cohort of genes (Fig. 1e). Considering only our filtered PAS-assigned reads ameliorated this bias, which led to a stronger correlation to relative mRNA expression levels (Pearson r = 0.88, Spearman rho = 0.85) (Fig. 1e). We speculated that the bias in the gene body method relative to the PAS-assigned read counting method could be

explained by the over-abundance of intron-mapping reads (Supplementary Fig. 1c) and enrichment of A-rich stretches that nucleate the production of internal priming artifacts. Indeed, a lasso regression model trained to predict the difference between the two strategies confirmed that intron length was strongly associated with inflated counts; moreover, "AAA" was the top-ranked of all 3-mers associated with inflated counts in the gene body (Pearson $r = 0.56$, Spearman rho $= 0.58$, Supplementary Fig. 2).

As a second QC, we evaluated the similarity between our IIR profiles to those derived from bulk 3P-seq data[8]. We considered the latter as a gold standard in accurately quantifying PAS abundances due to the involvement of a splint-ligation step in the 3P-seq protocol, which specifically removes internal priming artifacts[16]. We found that the IIR profiles for our PAS-assigned reads more strongly mirrored those of bulk data for two representative genes (Fig. 1f). Quantifying the deviation from bulk as the Mean Absolute Deviation (MAD) (Supplementary Fig. 3a) allowed us to measure the deviations between our pre- and post-processed data to bulk 3P-seq measurements. Applying this metric globally to all genes uncovered that 78% of genes exhibited improved similarity to bulk 3P-seq after the reads were assigned to PASs; moreover, 47% of genes achieved MAD ≤ 0.1 after read-to-PAS assignment, relative to only 7% of genes beforehand (Supplementary Fig. 3b). Inspection of IIR profiles for nine representative genes further confirmed the general improvement in consistency with bulk 3P-seq data (Supplementary Fig. 3c).

**Global differences in 3′-UTR length across mouse cell types and developmental time.** Having assigned reads to PASs and linked them to genes, we next sought to evaluate global properties of 3′-UTR shortening and lengthening (i.e., as quantified by differential PAS usage) across cell types and developmental time. Towards this goal, we computed a "gene by cell" sparse matrix of the mean length among all 3′-UTR isoforms, weighted by their respective counts. For each gene, we then computed each cell's deviation from the mean of 3′-UTR lengths across cells, considering only nonmissing values. Finally, for each cell, we computed the mean of these deviations across genes as a measure of the global behavior of the transcriptome through the perspective of APA. We projected these measurements onto a global map of 38 t-SNE clusters representing all major mouse cell types[31]. Highlighting the top-ten ranked t-SNE clusters with the largest differences, we discovered the greatest average 3′-UTR lengths among stromal cells and three neuronal cell types; in contrast, the shortest lengths occurred in three blood cell types, hepatocytes, chondrocytes, and osteoblasts (Fig. 2a). UMI counts were only weakly correlated to changes in 3′-UTR length and therefore were not a confounding variable (Supplementary Fig. 4a). Sub-clustering each of the 38 t-SNE clusters reinforced these findings but revealed additional heterogeneity within each cell type (Supplementary Fig. 4b). Segregating our dataset by the five sampled timepoints, we observed an apparent global 3′-UTR lengthening across developmental time (Fig. 2b). Finally, to quantify the joint impact of cell type and developmental stage, we computed the average behavior among genes and cells associated with each of 38 t-SNE clusters and 5 developmental stages. Partitioning the data in this manner reinforced our observation that the average 3′-UTR length increased in nearly every cell type as developmental time progressed (Fig. 2c). The underlying cell counts from our bins were not correlated to changes in 3′-UTR length for this analysis and therefore were not a confounding variable (Supplementary Fig. 4c). Neuronal cell types clustered together and exhibited the greatest 3′-UTR lengthening relative to

other clusters at E13.5; in contrast, blood cell types exhibited highly shortened 3′-UTRs at E9.5 and grew until E13.5 to mean lengths similar to those of other cell types at E9.5 (Fig. 2c).

Next, we evaluated differences in global 3′-UTR length with respect to developmental trajectories computed using UMAP, an embedding that more faithfully recapitulates cell–cell relationships and intermediate states of differentiation relative to t-SNE. Evaluating ten developmental UMAP trajectories[31], we again observed a global lengthening in 3′-UTRs in nearly every trajectory (Fig. 3a). Mirroring our previous findings, the neural tube/notochord and the neural crest trajectories (capturing neurons of the peripheral nervous system) showed the greatest lengths relative to other cell types at E13.5, while the hematopoiesis trajectory displayed the shortest lengths relative to other cell types at E9.5 (Fig. 3a). A visual comparison of these three trajectories with respect to changes in both developmental time and 3′-UTR length showed that the process of 3′-UTR lengthening occurred contemporaneously with cellular differentiation, with gradients of lengthening emerging in intermediate cellular states (Fig. 3b–d). Notably, in the hematopoiesis trajectory, a major difference in 3′-UTR length could be explained by the switch from primitive to definitive erythropoiesis, rather than gradual lengthening within either lineage (Fig. 3d).

**Dynamic gene-specific patterns of alternative polyadenylation across early development.** While our previous analyses revealed transcriptome-wide trends, it remained unclear how specific changes in APA manifested at the resolution of individual genes. To investigate this, we tabulated read counts assigned to each gene (i.e., for each PAS and developmental stage, aggregating information across cell types), and used a $\chi^2$ test[30] to evaluate statistically significant differences in APA for 8653 genes with at least 100 reads in each of the five developmental stages (Supplementary Data 2). This procedure identified 5169 genes surpassing a Benjamini–Hochberg (BH)-corrected false discovery rate (FDR) $P$ value threshold of 0.05. Evaluating the dynamics of the mean 3′-UTR length for this cohort of significant genes at each stage, we discovered that 62% of genes fell into a large cluster that exhibited consistent lengthening over time (Fig. 4a). While the majority of these genes showed the greatest increase in lengthening from E11.5 to E12.5, a minority lengthened the most strongly from E10.5 to E11.5 (Fig. 4a). About 38% of the significant genes did not simply lengthen across developmental stages, with about half of these progressively shortening over time (Fig. 4a). As an alternative method to evaluate transcriptome-wide changes, we computed the entropy across PASs for each gene and developmental stage. This alternative visualization scheme uncovered that ~75% of genes obey a progressive decrease in entropy, indicating that as developmental time progresses, a few PASs become increasingly dominant for the vast majority of genes; in contrast, ~15% of genes exhibited the opposite pattern of increased entropy across time, with the remaining displaying heterogeneous patterns (Supplementary Fig. 5a, b).

We extended our previous gene-centric IIR plotting scheme (Fig. 1d) to visualize the landscape of APA across the five developmental stages assayed, this time using a $\chi^2$ test to highlight individual PASs which were significantly different in at least one stage (Fig. 4b). Using this scheme, we visualized an assortment of genes from different clusters to dissect the nature of the isoform switching events contributing to changes in 3′-UTR lengths (Fig. 4c). Many of these genes contained dozens of PASs whose relative proportions significantly changed across time. For genes belonging to the dominant cluster (*Tmem33*, *Lrtm1*, *Dcp1b*, and *Add2* in Fig. 4a), later developmental stages led to the progressive selection of distal isoforms, leading to progressive 3′-UTR

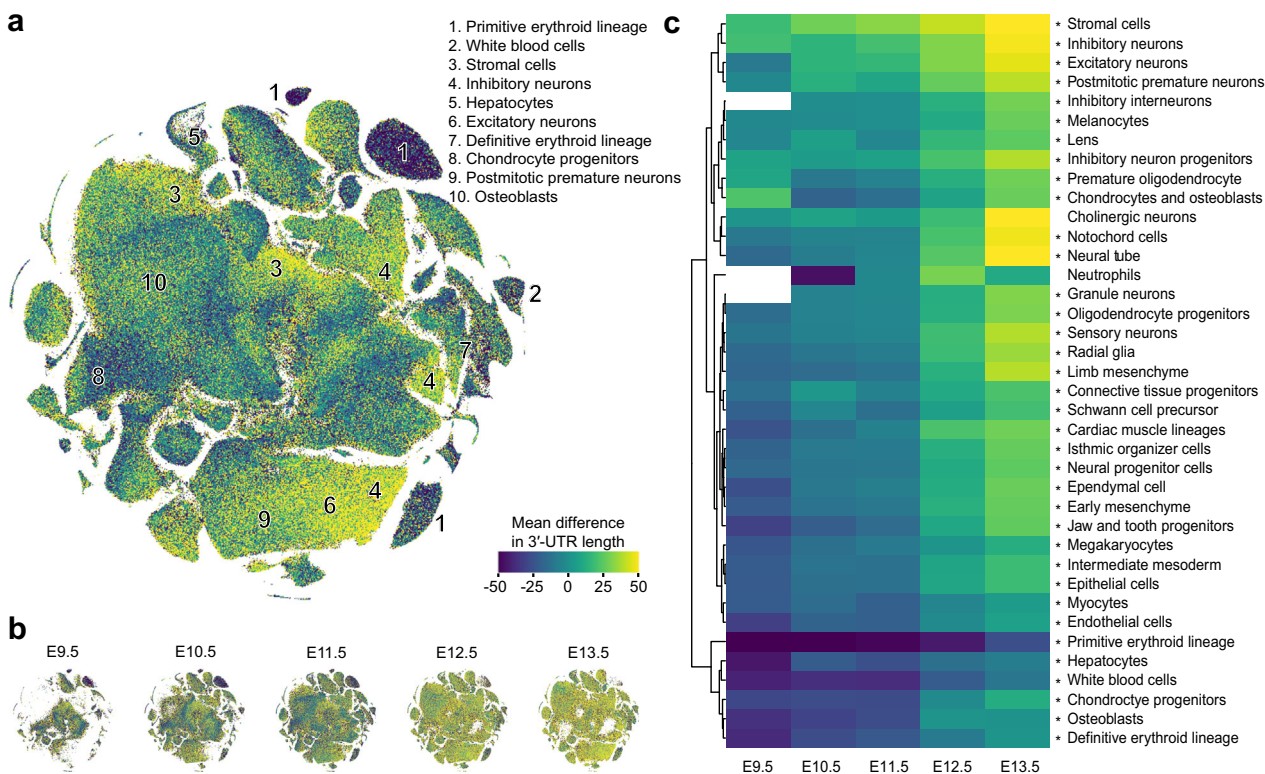

**Fig. 2 Differential 3′-UTR lengthening among diverse cell types and developmental stages. a** t-SNE embedding of all cells from all developmental stages sampled[31], with each cell colored according to the mean difference in 3′-UTR lengths across all genes. The top-ten ranked clusters with the greatest differences are annotated according to their corresponding cell type. **b** Shown are the same embeddings and color scales as those in (**a**), except after partitioning the dataset into its five composite developmental stages (spanning E9.5–E13.5). **c** Heatmap of the mean difference in 3′-UTR length after aggregating cells from each developmental stage and cell type, derived from each of 38 t-SNE clusters. Color scales are the same as those shown in (**a**). Missing values (shown in white) correspond to instances with too few (<20) cells to accurately estimate. Heatmap is clustered by Euclidean distance as a distance metric. Significant differences between the group means (i.e., for each row) were assessed by a one-way ANOVA test, with $P$ values adjusted for multiple hypothesis testing with a Bonferroni correction (*$P < 0.001$). Exact $P$ values are provided in Supplementary Table 1. See also Supplementary Fig. 4 for comparisons among t-SNE subclusters and an analysis of potential confounding biases.

lengthening (Fig. 4c). The opposite pattern was observed for a gene belonging to a smaller cluster (*Srl* in Fig. 4a), whereby the proximal isoform was selected more frequently over the distal as time progressed, leading to progressive 3′-UTR shortening (Fig. 4c). For yet other genes, the choice of distal isoforms was highly time-dependent. For example, *Mtap* displayed a near-complete proximal-to-distal isoform switching event in E11.5, subsequently lengthening beyond baseline levels in later developmental stages; in contrast, *Mrpl22* exhibited progressive shortening, with a dominant distal-to-proximal isoform switching event occurring in E10.5 (Fig. 4c).

Finally, we performed a similar gene-centric analysis, this time evaluating differences among individual cell types (i.e., aggregating information across the five developmental stages). Among 1491 genes with at least 20 reads in each of the 38 t-SNE clusters, we identified 1078 genes surpassing a BH-corrected FDR $P$ value threshold of 0.05, as evaluated by the $\chi^2$ test (Supplementary Data 4). This subset of significant genes largely clustered into four cell-type groups (C1–C4, Fig. 5a) when evaluating differences in mean 3′-UTR length, with cell types within each group displaying strongly correlated patterns across all of the genes. C1, which consisted primarily of neuronal cell types, was unique in that the vast majority of genes displayed global lengthening; conversely, the primitive erythroid lineage was dominated by genes experiencing 3′-UTR shortening (Fig. 5a). In special cases, we detected a highly gene-specific and cell-type-specific pattern, as in the case of *Bclaf1* showing 3′-UTR lengthening within a t-SNE

cluster annotated as lens cells (Fig. 5b). However, for most genes, all of the cell types within each cluster displayed a concerted shift toward either 3′-UTR lengthening (e.g., cluster C1 in *Gnb1*, C1 and C4 in *Samm50*) or 3′-UTR shortening (e.g., cluster C2 in *Polr3k*, C1 in *Hoxd4*) (Fig. 5b).

When visualizing PAS usage with respect to entropy, several cell types emerged as displaying interesting patterns: the primitive erythroid lineage showed heightened entropy across most genes, whereas neutrophils—and to a smaller degree, the lens—showed decreased entropy (Supplementary Fig. 6a, b). This observation suggests a potential for cell-type-specific regulatory mechanisms that guide a more stochastic or more defined choice of PASs, respectively. A smaller subset of genes, such as Sec11a, notably displayed higher entropy among neuronal cell types, consistent with an active mechanism governing a switch towards longer 3′-UTR isoforms (Supplementary Fig. 6a, b).

**Putative RNA-binding protein regulators of alternative poly-adenylation.** Reasoning that changes in the regulation of APA may be coupled to the dynamically changing expression of RNA-binding proteins (RBPs), we searched for RBPs with expression level differences across our five developmental stages and 38 cell types. Having demonstrated that PAS-mapping reads serve as an improved proxy for relative gene expression levels (Fig. 1e), we quantified gene expression levels for all protein-coding genes as counts per million (cpm) (Supplementary Data 6 and 7) and

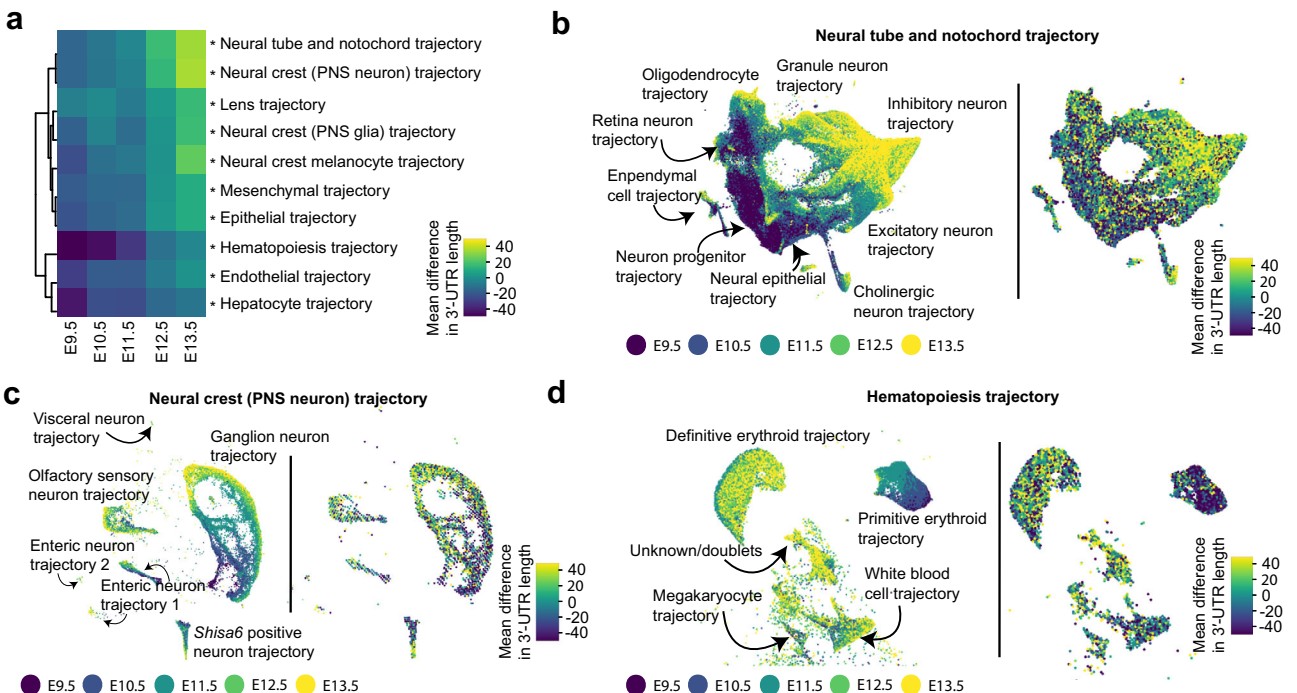

**Fig. 3 Differential 3′-UTR lengthening among diverse developmental trajectories. a** Heatmap of the mean difference in 3′-UTR length after aggregating cells from each developmental stage and one of ten developmental trajectories computed using UMAP[31]. Heatmap is clustered by Euclidean distance as a distance metric. Significant differences between the groups means (i.e., for each row) were assessed by a one-way ANOVA test, with P values adjusted for multiple hypothesis testing with a Bonferroni correction (*P < 0.001). Exact P values are provided in Supplementary Table 1. **b–d** UMAP embeddings of the neural tube and notochord (**b**), the neural crest of the peripheral nervous system (**c**), and hematopoiesis trajectory (**d**). Cells from each plot are colored by developmental stage (left panel) and mean difference in 3′-UTR length across genes (right panel). Mean differences are calculated with respect to the cells shown in the UMAP rather than all cells.

cross-referenced these genes to a database of putative RBPs in the mouse genome[38].

Evaluating relative differences in RBP expression across our five developmental stages, we observed that about 36% of RBPs exhibited increased expression across time (Supplementary Data 8; log₂ fold change >0 for E13.5 relative to E9.5), while the remainder decreased or were stable (Supplementary Fig. 7a). Performing a similar analysis across the 38 cell types (i.e., but independent of the developmental stage), we observed that about 26% of RBPs were enriched in neuronal lineages (Supplementary Data 8; log₂ fold change >0 for neuronal lineages relative to other cell types). Interestingly, the vast majority of RBPs were depleted in neuronal lineages relative to other cell types (Fig. 6a). Given our observation that 3′-UTRs lengthen both across the developmental stages and most dramatically in neurons (Fig. 2c), we sought to identify putative RBP regulators induced or repressed both across time and specifically in neurons relative to other cell types. We observed a statistically significant correlation in the RBP expression differences in these two conditions (Pearson r = 0.59), with a small group of outliers induced by at least twofold along both axes (Fig. 6b).

The most salient factors that were induced comprise a large family of ELAV-like (i.e., "Embryonic Lethal Abnormal Vision"-like) RBPs, including *Elavl2-4* (also known as *HuB*, *HuC*, and *HuD*, respectively) and *Celf2-6* (CUGBP Elav-like family members, also known as *BRUNOL-3*, *1*, *4*, *5*, and *6*, respectively) as well as splicing regulators *Nova1-2* and *Rbfox1-3*; in contrast, *Hnrnpf* and *Ptbp1*, which have been implicated in APA regulation, were among those depleted (Fig. 6b). Additional top-ranked RBPs (i.e., including those which are either strongly activated or repressed) also serve as candidate regulators of APA (Supplementary Data 8). The expression of *Elavl2-4*, *Rbfox1-3*,

and *Celf2-6* monotonically increased in expression across the developmental stages (Fig. 6c and Supplementary Fig. 7b, c), and were significantly higher in neuronal cell types, while *Elavl1* (also known as *HuR*) and *Celf1* (also known as *BRUNOL-2*) remained at similar levels in each context (Fig. 6d). In contrast, *Hnrnpf* and *Ptbp1* monotonically decreased (Supplementary Fig. 7d) and were relatively higher in other cell types relative to neurons (Fig. 6d). These expression patterns are broadly consistent with the known brain-specific and ubiquitous expression associated with ELAV-like[39–41] and Nova[42] family members, and support the growing functional evidence for *Elavl2-4*[43,44], *Rbfox2*[45], and *Nova1-2*[42] in the regulation of APA. In addition, we found that *ELAVL2-4*, *CELF2-6*, and *RBFOX1-3* form an experimentally supported network of protein-protein interactions[46] (Supplementary Fig. 7e). Our data are also consistent with the blood-enriched expression of *Hnrnpf* and *Ptbp1*, factors known to competitively bind with CstF-64 to promote proximal 3′-UTR isoform choice[47–49].

## Discussion

Despite the rapid growth of single-cell RNA sequencing data in recent years, the vast majority of analyses routinely overlook the phenomenon of alternative polyadenylation. Although scRNA-seq was initially developed to measure gene expression levels, multiple orthogonal forms of information are also effectively captured. For example, RNA velocity analysis, which estimates future transcriptome state by modeling intron/exon ratios, illustrates the ability to extract dynamical information about cellular differentiation[50]. In this work, we further develop a computational pipeline to quantify 3′-ends in scRNA-seq data by cross-referencing an integrated annotation set of 3′-UTRs and polyadenylation sites. This pipeline closely recapitulates prior bulk measurements, yet further enables a more granular understanding of APA with respect to both time and

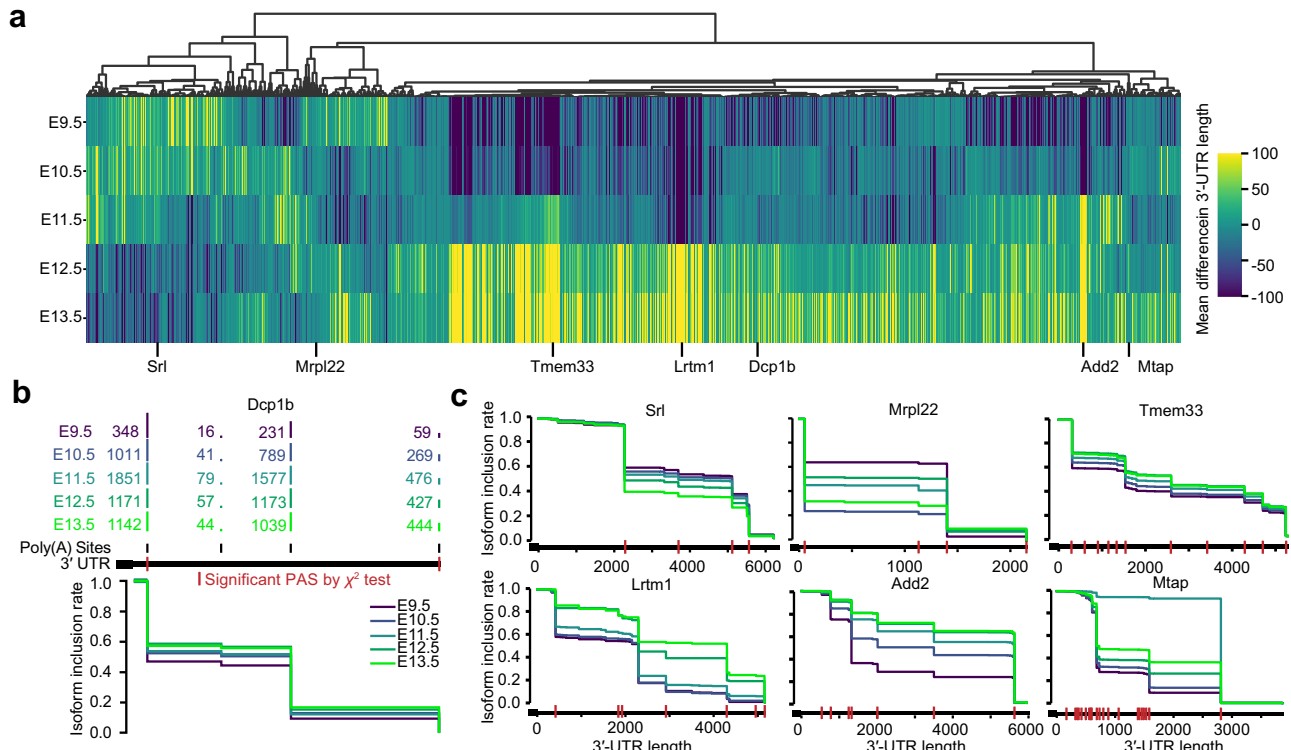

**Fig. 4 Identification of distinct gene lengthening patterns and genes responsible for overall data trends across embryonic ages. a** Heatmap of mean differences in 3′-UTR lengths for 5169 genes with significant differences in PAS usage across embryonic stages. Heatmap is column-centered and clustered by Pearson correlation as a distance metric. **b** Schematic of IIR plot visualization using PAS counts for each of five embryonic stages. Vertical red lines along the 3′-UTR indicate PASs that are significantly different between stages by the $\chi^2$ test ($P < 0.05$). **c** IIR plots for six genes among representative clusters shown in (**a**), and colored by embryonic stage. See also Supplementary Fig. 5 for clustering according to differences in entropy. See also Supplementary Data 3 for a table of read counts associated with each PAS for each gene and embryonic stage.

cell type. Although the utility of scRNA-seq to give insight into APA has been recognized recently[30], we extend this line of work to an expansive atlas of cell types in a developmental time course encompassing 61 embryos and spanning multiple stages of embryonic development[31].

Our findings reinforce the principle that the most proliferative cell types such as blood maintain shorter 3′-UTRs[11,12], on average, while lowly proliferative ones such as neurons maintain lengthened 3′-UTRs[17,23]. As differentiation progresses, cells of all types naturally become less proliferative, leading to an observed global lengthening of 3′-UTRs in all cell types[22,51]. A major functional consequence of this is that the global shortening of 3′-UTRs could lead to the evasion of microRNA-mediated repression, resulting in greater mRNA stabilities across the transcriptome and enhanced protein synthesis rates in proliferative cells[11,12]. In some cases, 3′-UTR shortening might also potentiate enhanced microRNA-mediated repression of anti-proliferative genes[52]. In contrast to previous work, which often binarized the landscape of 3′-termini into proximal and distal isoforms due to a limited PAS annotation set[22,23], we develop more general metrics (e.g., changes in mean length and entropy) that consider the relative proportions of the many PASs within each gene. While most genes obey a canonical pattern of 3′-UTR lengthening and a concomitant reduction in entropy over time, a small subset of genes deviate from this trend. While the former can be explained by modulation of *trans*-acting regulators of APA that disfavor weak proximal PASs over time, genes deviating from the trend can potentially be explained by several possible regulatory mechanisms: (i) enhanced recruitment of the cleavage and polyadenylation (CP) machinery to a proximal

PAS by a nearby RBP motif; (ii) suppressed post-transcriptional recruitment of CP machinery to distal PASs; or (iii) epigenetic regulation of PAS choice by CpG methylation status, nucleosome positioning, or histone modification state[1]. In addition, we observed that most cell types can be grouped into one of four clusters that obey similar trends across genes. Collectively, these observations are consistent with the evolution of regulatory mechanisms that act in a gene-specific and tissue-dependent manner[1,5].

An investigation into putative RNA-binding protein regulators that are co-activated in cellular contexts experiencing 3′-UTR lengthening revealed the induction of RBPs of the ELAV-like family, including *Elavl2-4* and *Celf2-6*, as well as splicing regulators *Nova1-2* and *Rbfox1-3*. Prior work provides functional evidence that fly orthologs of the ELAV-like induce neural-specific 3′-UTR lengthening through competition with CstF[43,44,53–55], and that mammalian *Elavl2-4* can also regulate APA[41,56]. Moreover, *Nova1-2*[42] and *Rbfox2*[45] have been directly implicated as regulators of APA in mouse and rat cells, respectively. Although only *Celf2* has been shown to regulate APA[57], evidence for the roles of *CELF* family proteins in this process include: (i) enriched expression in neurons and later developmental stages; (ii) direct interaction with *RBFOX* and *ELAVL* family members; (iii) nuclear localization[39]; (iv) enriched binding to the 3′-UTR terminus[58]; (v) interaction with U2 snRNP[39], which is known to promote distal isoform usage[59]; and (vi) roles in splicing[39,40]. Conversely, we also detect relative induction of *Ptbp1* and *Hnrnpf* in cellular contexts experiencing 3′-UTR shortening, which are also believed to compete with CstF-64 in the blood[47–49].

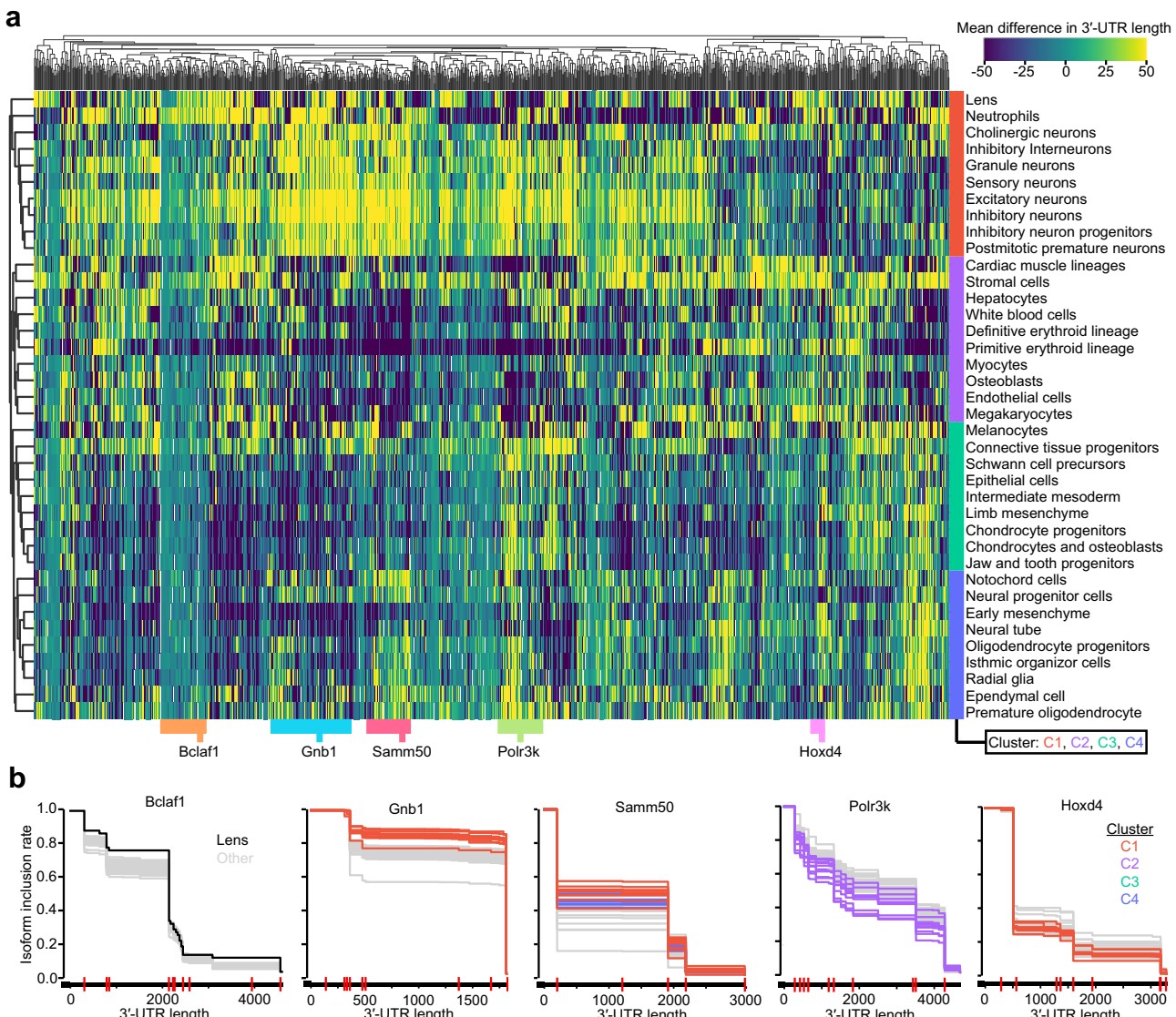

**Fig. 5 Identification of distinct gene lengthening patterns and genes responsible for overall data trends across embryonic ages. a** Heatmap of mean differences in 3′-UTR lengths for 1078 genes with significant differences in PAS usage across cell types derived from 38 t-SNE clusters. Heatmap is column-centered and clustered in both rows and columns by Pearson correlation as a distance metric. Row colors indicate the four dominant cell-type clusters. Column colors indicate five representative clusters chosen from the dendrogram above. **b** IIR plots for five genes among the representative clusters shown in (**a**), and colored by either individual cell types or cell-type clusters shown in (**a**). Gray lines allude to all other cell types. Vertical red lines along the 3′-UTR indicate PASs that are significantly different between cell types by the $\chi^2$ test ($P < 0.05$). See also Supplementary Fig. 6 for clustering according to differences in entropy. See also Supplementary Data 5 for a table of read counts associated with each PAS for each gene and cell type.

During the course of this work, we attempted an analysis to characterize motifs enriched in the vicinity of differentially expressed PASs. This analysis was largely inconclusive because it was highly sensitive to certain parameters such as: (i) the length of the input sequence window around each PAS, (ii) the threshold to consider a PAS differentially expressed, and (iii) a lower threshold for a gene to be considered sufficiently expressed. The most conclusive motif that emerged from this analysis was the well-characterized AAUAAA motif, with less robust support for a repetitive GU-rich element that matched the *CELF* recognition element.

Given the exaggerated 3′-UTR shortening and lengthening we observed in blood and neuronal cell types, respectively, it is interesting to consider how these observations might give insight into regulatory function in these tissues. In the blood, the bias towards the selection of proximal PASs is partially caused by the induction of CstF-64[60,61], which is associated with the G0 to S

phase transition cell active in proliferating cells[62]. The levels of CstF-64 and competing factors such as U1A[63], PTB[49], and hnRNP F[47,48] are thought to regulate the choice of low-affinity PASs associated with proximal 3′-UTR isoforms[64]. These mechanisms influence immunoglobulin (Ig) secretion in plasma cells[60,61,65] as well as isoform choice of FKBP12[66] and NF-ATc[67], key transcription factors which function in T cells. These isoform switching events enable plasma cells to perform their most fundamental function: secreting Ig[68].

In neurons, it has been previously observed that APA guides differential mRNA localization[69,70], and that APA itself is directly regulated by neural activity[71] such as long-term potentiation[72]. These findings open the possibility that APA might serve as an important process in guiding mRNAs to axons and dendrites, thereby modulating synaptic potential. One promising direction for this work is to use our APA atlas, and those derived from

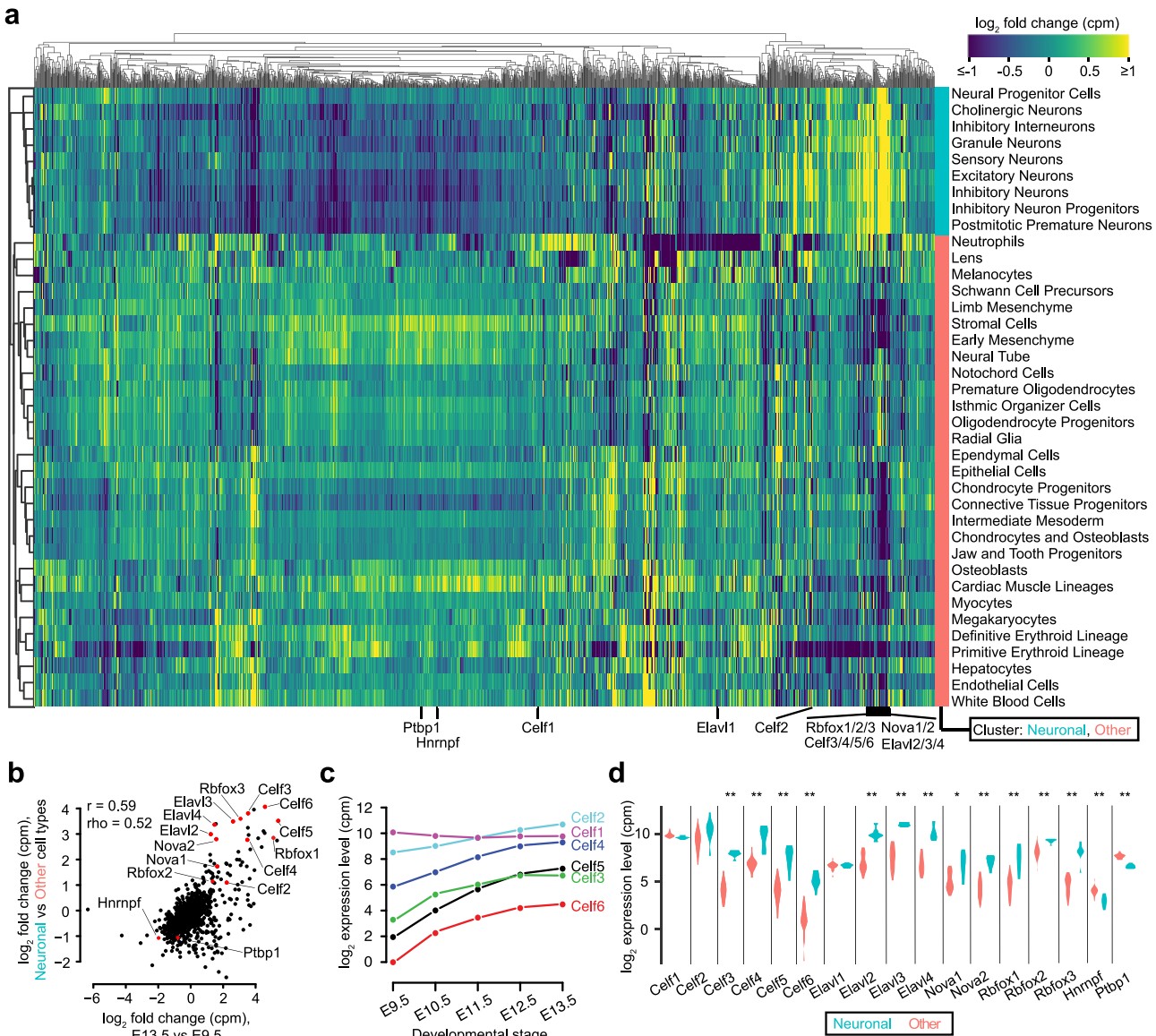

**Fig. 6 Evaluation of putative RNA-binding protein regulators of alternative polyadenylation. a** Heatmap of relative gene expression levels, quantified as log$_2$(counts per million), for a set of 1576 RBPs across cell types derived from 38 t-SNE clusters. Heatmap is column-centered and clustered in both rows and columns by Pearson correlation as a distance metric. **b** Relationship of changes in RBP expression between E13.5 vs. E9.5 relative to neuronal vs. other cell types. Also listed are the Pearson (*r*) and Spearman (rho) correlations. **c** Expression levels of *Celf1-6* across the five developmental stages. Expression is quantified in counts per million (cpm) and shown on a log$_2$ scale. All genes displayed significant differential expression across stages, with a Bonferroni-corrected *P* < 0.001 as assessed by the $\chi^2$ test for homogeneity, using PAS-mapping read counts associated with the RBP of interest versus all other RBPs. **d** Expression levels of genes highlighted in (**b**) in neuronal cell types relative to other cell types. Significant differences between the two groups were assessed by a two-sided Wilcoxon rank-sum test, with *P* values adjusted for multiple hypothesis testing with a Bonferroni correction (*$P$ < 0.01, **$P$ < 10$^{-4}$). Exact *P* values for panels (**c**, **d**) are provided in Supplementary Table 1.

other single-cell datasets[73], to further dissect how differential mRNA localization across neuronal subtypes might contribute to their functional specialization.

We anticipate that the general computational framework that we have developed herein will be broadly applicable toward assessing the landscape of APA in existing and future single-cell RNA-seq datasets from different organisms. Our characterization of APA across genes, cell types, and developmental stages of a mammalian organism will also serve as a resource to further guide the discovery of new regulatory mechanisms that control APA. Finally, it may help to dissect how these changes impact the function of mRNA with respect to its cellular localization, half-life, and translation in cell types throughout the body.

## Methods

**An integrated set of mouse 3′-UTRs.** We established an integrated set of mouse 3′-UTR annotations for protein-coding genes in which each unique stop codon was associated with a representative transcript with the longest annotated 3′-UTR[9], using the Gencode M25 "comprehensive" set[33] as our initial annotations (Supplementary Data 1). For each unique stop codon, we selected the longest 3′-UTR from three additional resources: (i) RefSeq (March 2020 release)[36], (ii) 3′-UTRs with extreme lengthening[23], using liftOver[74] to remap the coordinates from mm9 to mm10, and (iii) bulk 3P-seq-based annotations derived from mouse muscle, heart, liver, lung, kidney, brain, testes, and white adipose tissues as well as NIH 3T3 and mESC cell lines[8]. The choose_all_genes_for_TargetScan.pl Perl script in the TargetScanTools Github[35] was used to integrate these databases, allowing a 3P-seq read to exist up to 5,400 nt (i.e., the 99th percentile of annotated 3′-UTR lengths) downstream of a stop codon.

In certain scenarios, such as in the case of alternative splicing of the terminal exon, a gene is potentially associated with many unique stop codons, each of which

is associated with its own 3′-UTR annotation. We, therefore, sought to avoid a bias in which genes with many such transcript isoforms would be overrepresented in the downstream results, and to avoid the possibility that PASs would be counted redundantly in cases in which multiple different 3′-UTRs overlapped the same genomic coordinates. We, therefore, carried forward a single transcript isoform with the greatest number of 3′-UTR mapping reads (or a random top-ranked one in the case of a tie) to represent each gene. To perform this counting, scRNA-seq reads were intersected with our 3′-UTR annotation set using bedtools intersect (-wa -wb -s)[75].

**An integrated set of mouse poly(A) sites**. To generate our union PAS set, we integrated three PAS annotation databases: Gencode M25[33], PolyA_DB v3[20], and PolyASite 2.0[21]. First, PASs within ± 10 nt of another PAS within the same database were collapsed by selecting the most downstream PAS. Next, the following procedure was implemented to reduce redundancy between databases: (i) we collected PASs from PolyASite 2.0, (ii) we added PASs from PolyA_DB v3 not within ± 10 nt of the current PAS set, and (iii) we added PASs from Gencode M25 not within ± 10 nt of the current PAS set. This method of sequential addition led to a total of 164,772 PASs in our union set; we provide the genomic coordinates and corresponding read counts associated with this set (Supplementary Data 3 and 5).

**Calculation of 3′-UTR lengths, relative length differences, and corresponding visualizations**. Reads were mapped to the mm10 genome and collected from previous work[31] (GEO ID: GSE119945). Reads were then filtered according to their proximity to a known PAS. Associated motif analyses using DREME v5.0.5[76] and lasso regression models[77] demonstrated that these read filtering criterion improved the quantitation of 3′-UTR isoforms. 3′-UTR length corresponding to a given read (i.e., which remained after read filtering) was computed as the distance from the stop codon to the read's assigned PAS, minus the length of any intervening intron(s). These 3′-UTR lengths were used to compute a "gene by cell" sparse matrix of the mean length among all 3′-UTR isoforms, weighted by their respective counts. For each gene, we then computed each cell's deviation from the mean of 3′-UTR lengths across cells, considering only nonmissing values. For heatmaps, these deviation values were then averaged according to the labels assigned to each cell (i.e., with respect to t-SNE cluster, UMAP trajectory, and/or developmental stage). Cell labels were based upon those previously assigned[31]. When indicated in the legend, in some instances the heatmaps were further centered by subtracting the mean of the row or column. t-SNE plots were visualized using the hexbin (grid-size = 500, $v_{min} = -50$, $v_{max} = 50$) function from pyplot, which averages values from cells captured in local bins. UMAP plots were binned by splitting each of the $x$, $y$, and $z$ coordinates into 150 equally sized bins. For all of our analyses presented in Figs. 2 and 3, we experimented with testing increasingly stringent minimal read count thresholds to retain genes and/or cells in our sparse matrix, and achieved highly robust results independent of the thresholds selected.

**Gene-level isoform inclusion rate plots and corresponding statistics**. For each gene, we counted reads assigned to each PAS to build contingency tables of counts for either (PAS by developmental stage) (Supplementary Data 3) or (PAS by cell type) (Supplementary Data 5). We then computed statistical significance using the $\chi^2$ test for homogeneity as computed by the chisquare function in scipy, either with respect to the entire gene (Supplementary Data 2 and 4) or with respect to each PAS (axis = none or axis = 1, respectively). In both cases, we provided a matrix of expected counts, based on the joint probability of each cell multiplied with the total counts in the matrix. For the gene-level $\chi^2$ test, we further derived a Benjamini–Hochberg (BH) based $q$ value to account for the FDR. Considering the read counts associated with each PAS position, isoform inclusion rates were visualized in the same manner as previous work, which allude to this plotting style as the affected isoform ratio (AIR) plot[8,9,35]. Much like a survival curve, the IIR quantifies the proportion of 3′-UTR isoforms that include a given nucleotide position.

**Search for putative RBP regulators**. To evaluate changes in gene expression associated with RBPs, we first computed gene expression levels for all protein-coding genes. Toward this goal, we summed the counts associated with PAS-mapping reads for all unique PASs (i.e., as assessed by chromosomal coordinate) across all transcripts (i.e., including those with alternative last exons) corresponding to each gene, using our count tables partitioned either by developmental stage (Supplementary Data 3) or by cell type (Supplementary Data 5). Counts were then normalized by the stage or cell type into counts per million (cpm) (Supplementary Data 6 and 7) and then $\log_2$-transformed. Genes were annotated as an RBP if their gene name matched one of 1882 mouse genes annotated as a putative RBP[38]. For the subset of 1576 RBPs meeting an expression threshold of 2 cpm in at least one of the samples tested, we computed the fold change of each gene across time as [$\log_2$(cpm at E13.5) – $\log_2$(cpm at E9.5)] and in neurons relative to other cell types as [mean $\log_2$(cpm in neurons) – mean $\log_2$(cpm in other cell types)] (Supplementary Data 8), where neurons are defined as the cell types in the cluster shown in Fig. 6a.

**Reporting summary**. Further information on research design is available in the Nature Research Reporting Summary linked to this article.

## Data availability

The analyses presented in this study are based on publicly available data, including: RefSeq (March 2020 release)[36], 3′-UTRs with extreme lengthening[23], Gencode M25[33], PolyA_DB v3[20], PolyASite 2.0[21], bulk 3P-seq data[8], scRNA-seq data (GSE119945)[31], and a list of mouse RBPs[38].

## Code availability

The computational pipeline to reproduce the core results of this work is provided under the MIT open-access license at the following link: https://github.com/serenolopezdarwin/apanalysis.

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

## Acknowledgements

We thank Junyue Cao and other members of the Shendure Lab for discussions surrounding the single-cell data as well as Nimrod Rubenstein for critical commentary. We are grateful to Eric Lai for discussions surrounding RBP analysis and ELAV-related regulatory mechanisms. This work was supported by the Paul G. Allen Frontiers Foundation (Allen Discovery Center grant to J.S.), NIH grant R01HG010632 (to J.S.), and the NRSA fellowship 5T32HL007093 (to V.A.). J.S. is an investigator of the Howard Hughes Medical Institute.

## Author contributions

V.A. conceived of the study, designed analyses, and performed RBP analysis. S.L.-D. performed the remaining computational analyses and generated tables. V.A. and S.L.-D. generated figures and wrote the paper with feedback from D.R.K. and J.S.

## Competing interests

V.A. and D.R.K. are employees of Calico Life Sciences. The remaining authors declare no competing interests.

**Additional information**

