## [Peer Review File · Nature Communications]

The landscape of alternative polyadenylation in single cells of the developing mouse embryoREVIEWER COMMENTS

Reviewer #1 (Remarks to the Author):

Alternative polyadenylation is a widely used mechanism in post-transcriptional gene regulation. The resulting alternative 3'-UTR isoforms modulate diverse mRNA fate and function, including localization, stability and translation. This manuscript analyzed a collection of 2 million single-cell transcriptomes to achieve a transcriptome-wide and organism-wide map of APA landscape dynamics during mammalian embryo development. This study provide a comprehensive resource for cell type specific gene regulation through APA. No doubtly, this resource will be of great value for developmental studies. The analyses presented in the current manuscript are of good quality and the conclusions are convincing in general. I'd consider this manuscript a good candidate for Nature communications. I have several comments for further improvement of the manuscript.

1. This study use combination of three set of available PAS annotations. Since a very large comprehensive single-cell transcriptome data are included in this study, have the author considered to establish a new set of PAS annotation based on the huge scRNA-seq dataset used in the current study de novo? It is likely that more PAS sites can be discovered, especially for those highly cell-type specific PAS sites.

2. The authors analyzed putative RBPs involved in the cell-type specific and developmental timing specific APA regulation. How about cis-motifs that enriched in the UTR regions changed during lengthening or shortening of the UTRs? Performing such analysis of cis-elements will likely strength the manuscript greatly.

3. The RBPs induced both across time and specifically in neurons were analyzed in Figure 6. How about RBPs repressed both across time and specifically in neurons? It remains possible that some RBPs may facilitate the choice of proximal PAS sites in cells with short 3'-UTRs.

4. Statistic tests needed to be performed for Fig. 6C and Fig. S7B-C to claim "significant" change.

5. The method for grouping t-SNE clusters in Fig. 5A need to be described in the method part, or in figure legend.

Reviewer #2 (Remarks to the Author):

In this work, the authors develop methods to quantify transcriptome-wide changes in polyadenylation across 2 million cells spanning five stages of mouse development. Although this has been the subject of a few recent papers, the present work is far more comprehensive. My overall impression is that the biological insights are somewhat limited, but the methods will be generally useful to the community, and highlight additional types of data that can be extracted from single cell RNA-seq datasets. I do have the following major and minor concerns.

Major comments

- The authors should add analyses showing that their results cannot be explained by differences in cell quality (e.g. the number of detected UMIs per nucleus) or dataset quality (e.g. the number of nuclei per sample). Are there differences in cell quality or dataset quality between the different cell types, embryonic stages, or 3'-UTR length classes that could explain the trends that you observed (for example, if low quality cells/datasets tend to have shorter 3'-UTRs, etc).
- Along these lines, immune cells have low RNA content, while neurons have high RNA content. Could differences in RNA content explain your results? These differences are also reflected in cell quality metrics (but driven by non-technical factors).
- The authors should include more information about read and cell filtering steps. For example, it

appears that their method removes nearly 90% of all reads, reducing their total to 200 million reads. While the two QC comparisons suggest that these filtered reads contain many internal priming artifacts, how do we know they don't also contain a large number of unannotated APA sites? Do the filtered reads have any evidence of association with poly(A) sites (such as Figure S1A). How many of the filtered reads are associated with homopolymeric stretches?

- As a follow-up point, it seems that most nuclei would contain very few reads that pass their stringent filtering (i.e. 200M reads / 2M nuclei \sim 100 reads/nucleus). Can the authors please include more information on their nucleus quality filtering? Did they include nuclei even if they only had 10 mapped reads? How many nuclei were actually used in the final analysis?
- Many conclusions are only supported by point estimates of mean 3'-UTR lengths, without information on the underlying distributions. These conclusions should be bolstered with appropriate statistics. Overall, I had trouble understanding how many genes support each claim? Could these averages be distorted by high-abundance transcripts? Etc.
- In Figure 2, please include statistical tests showing 3'-UTR lengthening across cell types and developmental stages (panels A and C)
- In Figure 3, please include statistical tests showing 3'-UTR lengthening across developmental trajectories (panels A, B, C, and D)
- In Figures 3B,C,D, what are the arrows pointing to? Where do the trajectories start and end? Are the trajectories different clusters of cells, or different axes that have been calculated from the UMAP? Please clarify. It would be helpful to plot developmental stage against some measure of "pseudotime" to validate that these are indeed trajectories. For example, in Figure 3D, it is difficult to interpret the hematopoiesis trajectory. The early parts of the trajectory actually appear to be low quality cells rather than progenitors?
- The RBP analysis in Figure 6 is suggestive but not conclusive. Many such genes are markers of mature neurons, as might be expected. I wonder if there are other (more quantitative) ways of identifying APA regulators, such as LASSO across cell types and developmental stages or cell subtypes (this is more of a comment)

Minor comments

- In the future, it would be very helpful if the authors could use line numbers
- Because the authors are using a single nucleus dataset, they should consider changing "cells" to "nuclei" throughout the text (where appropriate)
- Would be nice to explain how the APA databases differ (1-2 sentences)
- The abbreviations "3' UTR" and "3'-UTR" are inconsistently used
- In the abstract please define RBP, ELAV, and all abbreviations in the text
- Are all FDRs calculated with B-H?
- In Figure 4BC, the significance is difficult to read
- Figure S1 needs a legend for ACGT. Inconvenient for legend text to refer to 1B.
- The labels in Figure S7D are difficult to read

Reviewer #3 (Remarks to the Author):

In this study, the authors analyzed more than 2 million single cells during E9.5-E13.5 of mouse embryo development stage by using single-cell RNA sequencing, and showed the overall quantification of the 3'-UTR in different cell types. They found that during the embryonic development, the hematopoietic lineage has a shorter 3'-UTR, and there is a longer 3'-UTR in neuronal cell types, and that the 3'-UTR undergoes global extension. The author successfully provides a transcriptome-wide of the dynamic landscape of variable polyadenylation during mammalian organogenesis. Overall, these findings are interesting, and the data analysis were well-designed and conducted with clear logic. However, the following comments need to be addressed before acceptance for publication on Nature Communication.

1. At the end of page 13, the author stated "that nearly 75% of genes obey a progressive decrease in entropy, and about 15% of genes exhibited the opposite pattern of increased entropy across time,

with the remaining displaying heterogeneous patterns". If possible, please explain the reasons for the heterogeneity of these genes in the discussion.

2. In the legend of Figure 5, how can the author judge that the selected genes can represent the overall data trend of the development of the entire embryo?

3. In the last paragraph on page 18, the author mentioned some percentages in Figure 6 about the trend of increase and decrease, and it is recommended that these percentages be added to Figure 6.

4. In the last paragraph on page 18, the author stated "We found a strong correlation in RBP expression differences (Pearson $r = 0.59$)", however, $R=0.59$ should not be interpreted as a strong correlation.

5. At the end of the first paragraph of the discussion section, "we extend this line of work to an expansive atlas of cell types in a developmental time course spanning multiple stages of embryonic development", how many embryos of all cells from the data analysis should be indicated in the right place, and whether there are also specificities of neuronal 3'-UTR extension and blood shorter 3'-UTR between different embryos?

6. In the second paragraph of the discussion section, "While most genes obey a canonical pattern of 3'-UTR lengthening over time, a small subset of genes deviate from this trend", the author should discuss it in detail so that readers can understand a small part more easily of the reason why genes deviate from this trend.

7. The author's pointed out that the increase in 3'-UTR length related to the development of neural cell lineage may be related to RBP, and an in-depth analysis of genes with log fold change >2 , but this part of the content still lacks the regulation of detailed discussion of the differential expressed genes on log fold change < -2 , and how the length of 3'-UTR regulates the binding of RBP needs to be clearly clarified, the potential significance of regulatory of the 3'-UTR length during the entire developmental process can be explored.

8. The author discusses the changing trend of 3'-UTR length in two different systems of neural and hematopoiesis during mouse embryonic development, and is supported by various analyses. However, at the end of the conclusion, the author seems to only discuss the potential regulatory role of APA in the nervous system while the hematopoietic system seems to be ignored, which needs to be discussed in detail.

POINT-BY-POINT RESPONSE TO REVIEWERS

We thank the three reviewers for their encouraging comments as well as valuable constructive criticism, which we believe has helped us to strengthen our manuscript. Please find below our detailed responses to each of the points raised alongside our revised manuscript. Reviewer comments are replicated in full in black, with our inline responses in blue.

Reviewer #1 (Remarks to the Author):

Alternative polyadenylation is a widely used mechanism in post-transcriptional gene regulation. The resulting alternative 3'-UTR isoforms modulate diverse mRNA fate and function, including localization, stability and translation. This manuscript analyzed a collection of 2 million single-cell transcriptomes to achieve a transcriptome-wide and organism-wide map of APA landscape dynamics during mammalian embryo development. This study provides a comprehensive resource for cell type specific gene regulation through APA. No doubt, this resource will be of great value for developmental studies. The analyses presented in the current manuscript are of good quality and the conclusions are convincing in general. I'd consider this manuscript a good candidate for Nature communications. I have several comments for further improvement of the manuscript.

1. This study use combination of three set of available PAS annotations. Since a very large comprehensive single-cell transcriptome data are included in this study, have the author considered to establish a new set of PAS annotation based on the huge scRNA-seq dataset used in the current study de novo? It is likely that more PAS sites can be discovered, especially for those highly cell-type specific PAS sites.

We agree that this idea is a natural approach to the problem. Indeed, when beginning this project, we initially sought to directly use the scRNA-seq data to annotate PASs. However, we quickly discovered that such an approach led to both a high false positive rate (e.g. due to internal priming artifacts) as well as a high false negative rate (e.g. due to the property that the reads are of variable length, often map several hundred nucleotides upstream of the true PAS, and frequently do not possess untemplated As, which are critical for PAS annotation). We therefore pivoted to the more conservative approach of building an integrative database of high-confidence PASs. To further address this comment and the related comment of Reviewer #2, point 3, we have added two additional figure panels demonstrating that the reads which were discarded are strongly associated with internal priming artifacts and are not associated with any known PAS signals, strongly reducing the likelihood that they capture novel PASs that were overlooked (please see expanded results and commentary below).

2. The authors analyzed putative RBPs involved in the cell-type specific and developmental timing specific APA regulation. How about cis-motifs that enriched in the UTR regions changed during lengthening or shortening of the UTRs? Performing such analysis of cis-elements will likely strength the manuscript greatly.

We agree that the discovery of *cis*-regulatory motifs involved in APA regulation is of crucial importance and utility for the field. Prior to manuscript submission, we invested substantial time to make headway into this problem, but ultimately concluded that the results were too speculative and lacked robustness to integrate into the manuscript with high confidence. Here we outline our methodology to evaluate this problem, the results we achieved, and our reasoning that guided us to this conclusion.

Methodology:

- i) For each of the neuronal vs non-neuronal conditions, as well as the E13.5 vs E9.5 condition, we first ranked all PASs according to their magnitude of up-regulation or down-regulation.
- ii) Next, we extracted the ± 100 nt window around the cut site for PASs corresponding to the top and bottom 10,000 PASs.
- iii) For each set of top and bottom 10,000 PASs, we generated a control set of PASs that were matched to our foreground set for proportion of PAS usage, but did not exhibit a change in the relevant conditions tested (*i.e.*, the neuronal vs non-neuronal condition or E13.5 vs E9.5 condition).
- iv) We searched for the unbiased differential enrichment of motifs with DREME using each pair of these foreground vs. background sets for each of the two conditions tested.
- v) We compared resulting motifs to a database of known RNA motifs using TOMTOM.
- vi) Steps i-v were repeated while varying certain parameters, such as testing larger sequence windows of ± 200 nt or ± 500 nt or more stringent ranking thresholds of the top 1,000 or 5,000 ranked PASs.

Results:

Evaluating motifs associated with increased PAS usage (corresponding predominantly to 3' UTR lengthening events), the top 2 motifs found were the AAUAAA and GU-rich motif, consistent with classically known PAS motifs:

While the literature has characterized the binding of CSTF to AAUAAA and CstF to the GU-rich motif, the latter also matched the CELF motif (found with a TOMTOM search as CUGBP1, an alternative name for the BRUNOL/CELF family of RBPs):

We also found motifs associated with down-regulated PASs (below), though these did not match any known factor by TOMTOM.

E13.5 vs E9.5 condition

Robustness testing and Conclusion:

Ultimately, we discovered that the results surrounding the GU-rich motif (our primary novel candidate) were unfortunately not robust, and we were also unable to reproduce a previously discovered U-rich motif (Seungjae Lee et al., *bioRxiv* 2020). In particular, we found that the entire motif analysis was highly sensitive to several parameters going into motif discovery: i) we must select a lower bound for gene expression to consider a PAS, ii) we must select a window around the PAS to evaluate enrichment (e.g., $\pm 100\text{nt}$), and iii) we must select how large of a change the specific PAS undergoes to be considered differentially up- or down-regulated (e.g., top 10,000 ranked PASs). While applying different thresholds to each of these parameters (e.g., a sequence window of $\pm 200\text{nt}$ or $\pm 500\text{nt}$ or top 1,000 or 5,000 ranked PASs) and performing our differential motif analysis on each pair of foreground and background sets, we discovered that the GU-rich motif seems highly sensitive to these parameters and was difficult to consistently reproduce. The only robustly reproducible motif associated with 3' UTR shortening and lengthening was the AAUAAA motif, which has been already well-characterized in the literature as being associated with distal 3' UTR isoforms. Therefore, the lack of robustness of our findings undermines our confidence for presenting this class of analysis.

To indicate this to the reader, we have now added the following passage in the discussion: “Although we attempted an analysis to characterize motifs enriched in the vicinity of differentially expressed PASs (data not shown), this analysis was largely inconclusive because it was highly sensitive to certain parameters such as: i) the length of the input sequence window around each PAS, ii) the threshold to consider a PAS differentially expressed, and iii) a lower threshold for a gene to be considered sufficiently expressed. The most conclusive motif that emerged from this analysis was the well-characterized AAUAAA motif, with less robust support for a repetitive GU-rich element that matched the *CELF* recognition element.”

3. The RBPs induced both across time and specifically in neurons were analyzed in Fig. 6. How about RBPs repressed both across time and specifically in neurons? It remains possible that some RBPs may facilitate the choice of proximal PAS sites in cells with short 3'-UTRs.

We concur that it is possible that RBPs which are repressed in neurons or across time could play a role in the choice of proximal vs distal PASs, a point which Reviewer #3 also makes. We have now examined those which were both induced and repressed, and have identified and integrated two additional candidates (*Hnrnpf* and *Ptbp1*) to the manuscript, which have precedent in the literature as being involved in APA regulation in the blood. We have also modified the following line to guide interested experimentalists towards better examining additional candidates: “Additional top-ranked RBPs (*i.e.*, including those which are either strongly activated or repressed) also serve as candidate regulators of APA (**Supplementary Table 8**)”.

4. Statistic tests needed to be performed for Fig. 6C and Fig. S7B-C to claim "significant" change.

We now report statistical tests on all of the RBPs shown in these figures and added the following line to both legends: “All genes displayed significant differential expression across stages, with a Bonferonni-corrected $p < 0.001$ as assessed by the χ^2 test for homogeneity, using PAS-mapping read counts associated to the RBP of interest versus all other RBPs.”

5. The method for grouping t-SNE clusters in Fig. 5A need to be described in the method part, or in figure legend.

The figure legend for **Fig. 5a** currently indicates how t-SNE clusters were grouped in the following line: “Heatmap is column-centered and clustered in both rows and columns by Pearson correlation as a distance metric.” Rows are shown as t-SNE clusters while columns are individual genes.

Reviewer #2 (Remarks to the Author):

In this work, the authors develop methods to quantify transcriptome-wide changes in polyadenylation across 2 million cells spanning five stages of mouse development. Although this has been the subject of a few recent papers, the present work is far more comprehensive. My overall impression is that the biological insights are somewhat limited, but the methods will be generally useful to the community, and highlight additional types of data that can be extracted from single cell RNA-seq datasets. I do have the following major and minor concerns.

Major comments

1) The authors should add analyses showing that their results cannot be explained by differences in cell quality (e.g. the number of detected UMIs per nucleus) or dataset quality (e.g. the number of nuclei per sample). Are there differences in cell quality or dataset quality between the different cell types, embryonic stages, or 3'-UTR length classes that could explain the trends that you observed (for example, if low quality cells/datasets tend to have shorter 3'-UTRs, etc).

We have added two panels (A and C) to the amended **Supplementary Fig. 4**, along with corresponding discussion in the main text. These panels demonstrate that (A) UMI count per

nucleus is very poorly correlated to changes in 3'-UTR length at the level of individual nuclei, and (C) nuclei count is not correlated to changes in 3'-UTR length at the level of developmental stage x cell type bins in **Fig. 2C**. Thus, neither UMI count nor nuclei count confound the analyses. These new panels are replicated below:

Supplementary Figure 4. Analysis of potential confounding biases and differential 3'-UTR lengthening among cellular subtypes. (A) Scatter plot showing the relationship between the number of reads, computed as unique molecular indexes (UMIs), relative to the mean change in 3'-UTR length among cells as shown in **Fig. 2A**. Regions are colored according to the density of data from grey and blue (low density) to yellow (high density). Also shown are the Pearson (r) and Spearman (ρ) correlations. (C) Scatter plot showing the relationship between the number of cells relative to the mean change in 3'-UTR length among developmental stage and cell type bins as shown in **Fig. 2C**. Also shown are the Pearson (r) and Spearman (ρ) correlations.

2) Along these lines, immune cells have low RNA content, while neurons have high RNA content. Could differences in RNA content explain your results? These differences are also reflected in cell quality metrics (but driven by non-technical factors).

The clearest way to directly evaluate this idea is to presume that RNA content in a cell is in turn related to UMI counts. Our results addressing the previous question demonstrate that UMI counts (serving as a proxy for RNA content) are not associated with average 3'-UTR lengthening and shortening in cells. Filtering steps were also taken to remove nuclei with an inordinate number of UMIs (see point 4 below).

3) The authors should include more information about read and cell filtering steps. For example, it appears that their method removes nearly 90% of all reads, reducing their total to 200 million reads. While the two QC comparisons suggest that these filtered reads contain many internal priming artifacts, how do we know they don't also contain a large number of unannotated APA sites? Do the filtered reads have any evidence of association with poly(A) sites (such as Fig. S1A). How many of the filtered reads are associated with homopolymeric stretches?

After randomly sampling 20,000 reads that were discarded by our filtering criteria, we performed the analysis suggested, akin to **Supplementary Fig. 1A**, and obtain the following result:

There is clear A-richness downstream (but not upstream) of the read 3'-terminus as expected.

Moreover, in the hypothetical scenario that these reads are associated with annotated PASs, they should be enriched for the AAUAAA PAS motif, expected upstream of the cut site. We have now run a DREME analysis to evaluate motif enrichment in the vicinity of such reads (in the same +/- 200nt window around the 3' termini of the reads, as shown above). The homopolymer AAAANAAA was the #1-ranked motif and strongly enriched in the vicinity of the reads as expected, occurring in 4958/20000 reads (24.8%):

However, no AAUAAA motif was detected as a significantly enriched k-mer among the other top-ranked motifs discovered. The primary reason for ~75% of reads not containing the 8-mer above is that there is also enrichment of shorter homopolymeric A stretches, as short as 3mers (**Supplementary Fig. 3B**), as well as other degenerate A-rich k-mers that are likely sufficient to seed the internal priming. As a positive control, the subset of reads passing our filters were indeed enriched in the #1-ranked motif AAUAAA (E-value = 1e-1408) when performing the same analysis:

This analysis has now been integrated into the manuscript as **Supplementary Fig. 1E-F** with associated text in the main body to indicate that, by and large, the discarded reads are not

associated with cryptic PASs outside of the scope of our unified PAS database. We also note that in the scenario in which there were real PASs among the discarded reads, we would need to design a strategy to specifically isolate them away from background noise, which is in itself a separate research problem outside of the scope of our work.

4) As a follow-up point, it seems that most nuclei would contain very few reads that pass their stringent filtering (i.e. 200M reads / 2M nuclei ~ 100 reads/nucleus). Can the authors please include more information on their nucleus quality filtering? Did they include nuclei even if they only had 10 mapped reads? How many nuclei were actually used in the final analysis?

In this study, we evaluated the subset of nuclei retained by Cao et al, *Nature* 2019, whose methods describe the following: “Cells with fewer than 200 UMIs or over 3,172 UMIs (two standard deviations above the mean UMI count) were discarded”. Aside from our single nucleus figures such as **Fig. 2A**, most of the figures pool information from all valid nuclei, even if they contribute few reads to the final read count. As described in the methods: “For each gene, we then computed each cell’s deviation from the mean of 3’-UTR lengths across cells, considering only non-missing values”. Thus, cells with a zero read count will not have been considered in the calculation of the mean 3’-UTR length change for the gene of interest. In preparing our study, we also experimented with completely filtering out either genes and/or cells below a certain UMI count, and found highly similar results for all of the resulting figures. To show an example of this, here we show six different UMI thresholds and the corresponding results (as comparable to **Fig. 2B**) across the five developmental stages:

Only considering nuclei with >10 UMIs

Only considering nuclei with >100 UMIs

Only considering genes with >1,000 UMIs across all nuclei

Only considering genes with >10,000 UMIs across all nuclei

Only considering genes with >1,000 UMIs across all nuclei and nuclei with >100 UMI

Only considering genes with >10,000 UMIs across all nuclei and nuclei with >100 UMI

Increasing stringency on the minimum acceptable nuclei UMI counts led to severe cell loss across cell types and developmental stages. Thus, we opted to retain all cells, even those w/ few UMIs, for the most straightforward methodology that enabled us to retain the greatest number of counts after pooling the cells together by cell type and/or developmental stage.

We have added the following sentence to the methods to reflect this: “For all of our analyses presented in **Figs. 2-3**, we experimented with testing increasingly stringent minimal read count thresholds to retain genes and/or cells in our sparse matrix, and achieved highly robust results independent of the thresholds selected (data not shown).”.

Overall, 2,026,641 nuclei containing a non-zero UMI count were used in the final analysis.

5) Many conclusions are only supported by point estimates of mean 3'-UTR lengths, without information on the underlying distributions. These conclusions should be bolstered with appropriate statistics. Overall, I had trouble understanding how many genes support each claim? Could these averages be distorted by high-abundance transcripts? Etc.

We have added information regarding the statistics requested in points 6 and 7 below, which now uses information from the underlying distributions from individual nuclei. Each nucleus shown in **Fig. 2A-B** and **Fig. 3B-D** displays information from the average among 18,642 genes. Each gene receives equal weight in that the average 3'-UTR length change of a highly and lowly expressed gene is weighted equally. Using more stringent gene expression thresholds leads to results which are robust to threshold choice (as detailed in point 4 above).

6) In Figure 2, please include statistical tests showing 3'-UTR lengthening across cell types and developmental stages (panels A and C)

We have now added appropriate statistics to show significant lengthening to **Fig. 2B** and **Fig. 2C** (a one-way ANOVA test with Bonferroni-corrected p-values).

7) In Figure 3, please include statistical tests showing 3'-UTR lengthening across developmental trajectories (panels A, B, C, and D)

We have now added appropriate statistics to show significant lengthening in **Fig. 3A** (an ANOVA test with Bonferroni-corrected p-values). **Figs. 3B-C** only visualize the underlying data shown in **Fig. 3A** and so would have identical statistics in showing the relationship between developmental time and changes in 3'-UTR length.

8) In Figures 3B,C,D, what are the arrows pointing to? Where do the trajectories start and end? Are the trajectories different clusters of cells, or different axes that have been calculated from the UMAP? Please clarify. It would be helpful to plot developmental stage against some measure of “pseudotime” to validate that these are indeed trajectories. For example, in Figure 3D, it is difficult to interpret the hematopoiesis trajectory. The early parts of the trajectory actually appear to be low quality cells rather than progenitors?

The arrows simply serve the purpose of showing the group of cells corresponding to a particular subtrajectory within a general trajectory (e.g. Megakaryocyte subtrajectory from the hematopoiesis main trajectory). These trajectories and arrows are the same as those presented in Cao et al, *Nature* 2019, as we cite in our figure legend. Because they are colored by developmental stage (*i.e.*, E9.5 to E13.5), the axis along which the trajectory progresses is clear by the coloration of the figure (purple to yellow), and allude to different clusters of cells. We added the following sentence to the legend to further clarify these points: “Arrows point to the corresponding population of cells associated with the indicated subtrajectory, with pseudotime axes associated with developmental stage (Cao et al 2019).”

The boundaries of each trajectory are admittedly difficult to perceive but we feel showing them would make an already crowded figure even more crowded, and is slightly satellite to the key underlying point that there is a strong correspondence between the developmental stage and 3'-UTR lengthening (which now has statistical support for each trajectory as shown in the revised **Fig. 3A**).

With respect to the relationship between pseudotime and developmental time, Cao et al has already generated several figures and written a discussion showing the relationship between the two in the following excerpt from the study: “Although Monocle 3 did not have access to these labels, the sub-trajectories are highly consistent with developmental time (that is, cells ordered from E9.5 to E13.5; Extended Data Figs. 9, 10). To orient sub-trajectories, we identified one or several starting points as focal concentrations of E9.5 cells and then computed developmental pseudotime for cells present along various paths (Extended Data Fig. 11, Methods).”

Thus, the sub-trajectories themselves were in part defined by an enriched presence of early-stage nuclei, leading to a definitional association between pseudotime and developmental stage. For posterity, we can confirm that the two are indeed positively associated with one another after fitting a loess regression line between each. As an example, the “Neural tube and notochord” trajectories show a positive association between pseudotime and both embryonic stage (left panel) as well as changes in 3'-UTR length (right panel):

9) The RBP analysis in Figure 6 is suggestive but not conclusive. Many such genes are markers of mature neurons, as might be expected. I wonder if there are other (more quantitative) ways of identifying APA regulators, such as LASSO across cell types and developmental stages or cell subtypes (this is more of a comment)

Our primary purpose in this figure was to tie our results to a mechanistic understanding of APA regulation found in the literature, and to provide experimentalists with a starting point towards the identification of novel candidate RBP regulators. The heavy multicollinearity among genes shown in **Fig. 6A** would make a regression model such as lasso difficult to interpret, so we opted simply to show how all RBPs change and which ones are induced or repressed in cellular contexts exhibiting 3'-UTR lengthening. During our exploration of this problem, we also experimented with looking more deeply into the correlation between RBP expression and change in 3'-UTR length across cell types and developmental stages. However, such analysis led to a unimodal distribution in which many RBPs were strongly correlated, including those with relatively low expression level, making it difficult to isolate likely causal regulators from background noise.

Minor comments

- In the future, it would be very helpful if the authors could use line numbers

We have now added line numbers in our revision.

- Because the authors are using a single nucleus dataset, they should consider changing “cells” to “nuclei” throughout the text (where appropriate)

We see the point and this has been challenging in the past because different subsets of the field have different views about terminology. We changed from cells to nuclei in the abstract and introduction, but more generally have stuck with cells. For clarity, we added the following text: “Please note that in the remainder of the manuscript, we often use the term “cells” in relation to scRNA-seq profiles. However, all MOCA data was derived from nuclei rather than cells.”

- Would be nice to explain how the APA databases differ (1-2 sentences)

We have now added the following line in the main text body for further clarification: “The three databases differ in their use of manually curated annotation and Expressed Sequence Tag (EST) data (as in Gencode M25), amount of 3'-end sequencing sequencing data (246 and 178 mouse samples for PolyA_DB v3 and PolyASite 2.0, respectively), and computational processing pipelines.”

- The abbreviations “3' UTR” and “3'-UTR” are inconsistently used

Our reasoning for this was to use “3'-UTR” when contextually used as an adjective and “3' UTR” when it is used as a noun.

- In the abstract please define RBP, ELAV, and all abbreviations in the text

We have now revised the abstract with RBP accordingly. Because it is wordy to write “Embryonic Lethal Abnormal Vision” within the word limit of the abstract, we have expanded the acronym in the main body of the text instead.

- Are all FDRs calculated with B-H?

In our initial manuscript, the FDRs provided were B-H corrected unless explicitly indicated in the Figure legend (i.e., **Fig. 1D**). To address Reviewer 1’s comment on Point 4 and your comments on points 6 and 7, we have additionally added analyses that use Bonferroni corrections, and have noted so accordingly in the relevant legends.

While it was indicated in the Methods before, to make it a bit more explicit in the main body of the text, we have added the following modified lines in the revision.

“This procedure identified 5,169 genes surpassing a Benjamini-Hochberg (BH) corrected False Discovery Rate (FDR) p-value threshold of 0.05.”

“Among 1,491 genes with at least 20 reads in each of the 38 t-SNE clusters, we identified 1,078 genes surpassing a BH-corrected FDR p-value threshold of 0.05, as evaluated by the χ^2 test (**Supplementary Table 4**).”

- In Figure 4BC, the significance is difficult to read

Thank you for pointing this out. We have now made the red significance value lines thinner and longer in **Figs. 4BC, S5B, 5B, and S6B**, so that they are more easily visible.

- Figure S1 needs a legend for ACGT. Inconvenient for legend text to refer to 1B.

There is already an ACGT legend at the top right of each subplot of **Fig. S1**, so we’re not sure if we follow. The legend text refers to **Fig. 1B** to guide the reader at the relatedness between the two figures.

- The labels in Figure S7D are difficult to read

We have now substantially enlarged the size of **Fig. S7D** to make it easier to read the gene names.

Reviewer #3 (Remarks to the Author):

In this study, the authors analyzed more than 2 million single cells during E9.5-E13.5 of mouse embryo development stage by using single-cell RNA sequencing, and showed the overall quantification of the 3'-UTR in different cell types. They found that during the embryonic development, the hematopoietic lineage has a shorter 3'-UTR, and there is a longer 3'-UTR in neuronal cell types, and that the 3'-UTR undergoes global extension. The author successfully provides a transcriptome-wide of the dynamic landscape of variable polyadenylation during mammalian organogenesis. Overall, these findings are interesting, and the data analysis were well-designed and conducted with clear logic. However, the following comments need to be addressed before acceptance for publication on Nature Communication.

1. At the end of page 13, the author stated "that nearly 75% of genes obey a progressive decrease in entropy, and about 15% of genes exhibited the opposite pattern of increased entropy across time, with the remaining displaying heterogeneous patterns". If possible, please explain the reasons for the heterogeneity of these genes in the discussion.

The reasons for these changes in entropy are intertwined with changes in 3'-UTR length. We have changed the language in our discussion to the following: "While most genes obey a canonical pattern of 3'-UTR lengthening and a concomitant reduction in entropy over time, a small subset of genes deviate from this trend", and then expanded the discussion of the corresponding regulatory mechanisms that explain this effect. Please see our response to point 6 below for further details.

2. In the legend of Figure 5, how can the author judge that the selected genes can represent the overall data trend of the development of the entire embryo?

Our intention here was to unbiasedly show the entire set of genes that demonstrate a change in APA (**Fig. 5A**) to indicate the overall data trend. We then select large clusters that obey a certain cell-type specific behavior, and then sample one gene from each cluster to provide the reader with a sense of how changes in APA manifest at the level of individual genes within that cluster. Thus, the selected genes aren't meant to show the overall trend but only the trend of the specific cluster selected within the context of all genes that exhibit a significant change in APA.

3. In the last paragraph on page 18, the author mentioned some percentages in Figure 6 about the trend of increase and decrease, and it is recommended that these percentages be added to Figure 6.

We agree that the provenance of these percentages was unclear in the original draft. We have now substantially reorganized the paragraph to make their origin clear, with a link to the specific table from which they can be easily computed:

"Evaluating relative differences in RBP expression across our five developmental stages, we observed that about 36% of RBPs exhibited increased expression across time (**Supplementary**

Table 8, \log_2 fold-change > 0 for E13.5 relative to E9.5), while the remaining ones decreased or were stable (**Supplementary Fig. 7A**). Performing a similar analysis across our 38 cell types (*i.e.*, but independent of the developmental stage), we observed that about 26% of RBPs were enriched in neuronal lineages (**Supplementary Table 8**, \log_2 fold-change > 0 for neuronal lineages relative to other cell types)."

4. In the last paragraph on page 18, the author stated "We found a strong correlation in RBP expression differences (Pearson $r = 0.59$)", however, $R=0.59$ should not be interpreted as a strong correlation.

How one perceives the strength of a correlation is admittedly subjective. We have now changed the language to use more modest language for this line: "We observed a statistically significant correlation in the RBP expression differences (Pearson $r = 0.59$)".

5. At the end of the first paragraph of the discussion section, "we extend this line of work to an expansive atlas of cell types in a developmental time course spanning multiple stages of embryonic development", how many embryos of all cells from the data analysis should be indicated in the right place, and whether there are also specificities of neuronal 3'-UTR extension and blood shorter 3'-UTR between different embryos?

We have now modified the sentence to indicate the number of embryos used: "...we extend this line of work to an expansive atlas of cell types in a developmental time course encompassing 61 embryos and spanning multiple stages of embryonic development". Our entire analysis pooled read information from all embryos to have sufficient read depth to enable analyses for cell types such as neurons and blood, so we were only powered to make statements about average properties.

6. In the second paragraph of the discussion section, "While most genes obey a canonical pattern of 3'-UTR lengthening over time, a small subset of genes deviate from this trend", the author should discuss it in detail so that readers can understand a small part more easily of the reason why genes deviate from this trend.

We have now expanded the discussion to better describe possible reasons why genes deviate from this trend with the following text: "While most genes obey a canonical pattern of 3'-UTR lengthening and a concomitant reduction in entropy over time, a small subset of genes deviate from this trend. While the former can be explained by modulation of *trans*-acting regulators of APA that disfavor weak proximal PASs over time, genes deviating from the trend can potentially be explained by several possible regulatory mechanisms: i) enhanced recruitment of the cleavage and polyadenylation (CP) machinery to a proximal PAS by a nearby RBP motif, ii) suppressed post-transcriptional recruitment of CP machinery to distal PASs, or iii) epigenetic regulation of PAS choice by CpG methylation status, nucleosome positioning, or histone modification state (Di Giammartino et al. 2011)."

7. The author's pointed out that the increase in 3'-UTR length related to the development of neural cell lineage may be related to RBP, and an in-depth analysis of genes with log fold change >2, but this part of the content still lacks the regulation of detailed discussion of the differential expressed genes on log fold change < -2, and how the length of 3'-UTR regulates the binding of RBP needs to be clearly clarified, the potential significance of regulatory of the 3'-UTR length during the entire developmental process can be explored.

We concur that it is possible that RBPs which are repressed in neurons or across time could play a role in the choice of proximal vs distal PASs, a point which Reviewer #1 also makes. We have now examined those which were both induced and repressed, and have identified and integrated two additional candidates (*Hnrnpf* and *Ptbp1*) to the manuscript, which have precedent in the literature as being involved in APA regulation in the blood. We have also modified the following line to guide interested experimentalists towards better examining additional candidates: "Additional top-ranked RBPs (*i.e.*, including those which are either strongly activated or repressed) also serve as candidate regulators of APA (**Supplementary Table 8**)".

In addition, we have expanded the discussion about how modified 3' UTR regulation by RBPs play functionally significant roles in both neuronal and blood lineages (see comment 8 below).

8. The author discusses the changing trend of 3'-UTR length in two different systems of neural and hematopoiesis during mouse embryonic development, and is supported by various analyses. However, at the end of the conclusion, the author seems to only discuss the potential regulatory role of APA in the nervous system while the hematopoietic system seems to be ignored, which needs to be discussed in detail.

We agree that it would improve the manuscript to better emphasize the hematopoietic system in the discussion. In our revision, we have substantially expanded our discussion by adding a new paragraph that contextualizes our findings within the scope of what is known with respect to mechanism and function in the blood APA literature:

"In the blood, the bias towards the selection of proximal PASs is partially caused by the induction of CstF-64, which is associated with the G0 to S phase transition cell active in proliferating cells. The levels of CstF-64 and competing factors such as U1A, PTB, and hnRNP F are thought to regulate the choice of low affinity PASs associated with proximal 3'-UTR isoforms. These mechanisms influence immunoglobulin (Ig) secretion in plasma cells as well as isoform choice of FKBP12 and NF-ATc, key transcription factors which function in T cells. These isoform switching events enable plasma cells to perform their most fundamental function: secreting Ig."

REVIEWERS' COMMENTS

Reviewer #1 (Remarks to the Author):

My concerns have been adequately addressed.

Reviewer #2 (Remarks to the Author):

In the revised manuscript, the authors have addressed most major and minor concerns. They have added new analyses related to nucleus quality, data filtering, developmental trajectories, and have supported their work with the appropriate statistics. The manuscript and its methodology are sound and I support publication.

Reviewer #3 (Remarks to the Author):

The revised manuscript is significantly improved, all the previous concerns have been properly addressed. Thus I advise to accept for publication.